# Comparisons of Different Carbohydrate Quality Indices for Risk of Type 2 Diabetes in the Malmö Diet and Cancer Study

**DOI:** 10.3390/nu15183870

**Published:** 2023-09-05

**Authors:** Michaela Ramstedt, Suzanne Janzi, Kjell Olsson, Esther González-Padilla, Stina Ramne, Yan Borné, Ulrika Ericson, Emily Sonestedt

**Affiliations:** 1Nutritional Epidemiology, Department of Clinical Sciences Malmö, Lund University, SE-214 28 Malmö, Swedensuzanne.janzi@med.lu.se (S.J.); kjell.olsson@med.lu.se (K.O.); esther.gonzalez_padilla@med.lu.se (E.G.-P.); stina.ramne@med.lu.se (S.R.); yan.borne@med.lu.se (Y.B.); 2Diabetes and Cardiovascular Disease—Genetic Epidemiology, Department of Clinical Sciences Malmö, Lund University, SE-214 28 Malmö, Sweden; ulrika.ericson@med.lu.se

**Keywords:** type 2 diabetes, carbohydrates, free sugar, fiber, cohort

## Abstract

Carbohydrate quality might be more important than quantity to reduce type 2 diabetes (T2D) risk. Various metrics of carbohydrate quality exist; however, their associations with T2D have only been studied to a limited extent. Consequently, the aim was to investigate the association between four different pre-defined carbohydrate quality indices, with various amounts of fiber (≥1 g) and free sugar (<1 or <2 g) per 10 g of carbohydrates, and T2D risk among 26,622 individuals without diabetes from the Malmö Diet and Cancer cohort. Dietary data were collected through a food diary, diet frequency questionnaire, and interview. After a mean follow-up of 18 years, 4046 cases were identified through registers. After adjusting for potential confounders, no statistically significant associations were found for any of the indices. When excluding individuals with past dietary changes and potential misreporting of energy (36% of the population), lower risk was found for the following intake ratios: 10:1:2 carbohydrate:fiber:free sugar (HR = 0.82; 95% CI = 0.70–0.97), and 10:1&1:2 carbohydrate:fiber and fiber:free sugar, respectively (HR = 0.84; 95% CI = 0.72–0.97). Our findings indicate that adherence to a diet with high amounts of fiber and moderate amounts of free sugar in relation to total carbohydrate intake may be associated with a lower risk of T2D.

## 1. Introduction

Type 2 diabetes (T2D) is a chronic disease, and with its accompanying complications, such as cardiovascular disease, diabetic neuropathy, retinopathy, and nephropathy, it contributes to a wide range of public health problems around the world [1]. Today, over 400 million people have T2D [2]. T2D is characterized by prolonged hyperglycemia and an impaired metabolism of carbohydrates, proteins, and lipids [1]. Epidemiological evidence has shown that obesity is a major risk factor for T2D [1]. Other known risk factors include genetics, environmental factors, age, and lifestyle factors [3] such as diet, smoking, alcohol consumption, and a sedentary lifestyle [1,4,5]. Among the dietary factors, a higher intake of fiber, whole grains, legumes, fruits, vegetables, and coffee has been shown to be beneficial for the prevention of T2D, whereas a diet high in refined carbohydrates, sugar-sweetened beverages, processed meat, and alcohol has been associated with higher risk [1,6]. Carbohydrates in the diet have been thought to contribute to obesity, T2D, and other cardiometabolic diseases [7], but the relative proportion of total carbohydrate intake does not influence T2D noticeably [8]. Evidence from systematic reviews and meta-analyses suggests that carbohydrate quality might be more important than carbohydrate quantity in the reduction in cardiometabolic risk factors and incidence of T2D [7,9]. There are various markers of carbohydrate quality, including the content of dietary fiber, sugar, glycemic index, and glycemic load. Dietary fiber includes non-digestible carbohydrates and lignins that are intact and intrinsic in plants and are not absorbed in the small intestine of humans but may be fermented by bacteria in the colon [10]. Furthermore, since dietary fiber has been associated with a lower incidence of various diseases, such as T2D [7,11], the American Heart Association suggests an intake of at least 25 g per day [12], whereas the Nordic Nutrition Recommendations suggests an intake of at least 25–35 g/day [13]. It is debated whether the intake of sugars contributes to the development of T2D, and it is proposed that an excess calorie intake as a result of an increased sugar intake may be the causal factor [14]. International dietary guidelines from the World Health Organization suggest that people should limit their free sugar intake to less than 10% of their total energy intake (E%), ideally less than 5E% [15]. Free sugars include all mono- and disaccharides, except for those naturally present in whole foods that are intact, dried, or cooked, i.e., all sugars added during the manufacturing or cooking process, as well as sugars that are naturally occurring in fruit and vegetables that are no longer intact due to juicing or pureeing [16].

Since numerous evidence indicates that dietary fiber intake should be encouraged, and free sugar intake limited, new metrics of carbohydrate quality have been proposed, expressed as ratios of total carbohydrate and/or starches to dietary fiber content [11]. For example, based on the carbohydrate-to-fiber ratio in whole wheat, the American Heart Association has developed a ratio and established recommendations to select foods with at least 1 g of fiber per 10 g of total carbohydrate (<10:1 carbohydrate:fiber) [17]. Three additional metrics have been implemented to account for the recommended maximum intake of free sugars of <10E% or <5E%, as recommended by the World Health Organization, as well as the recommendation to consume at least 25 g fiber/day, as recommended by the American Heart Association [11,18]. These metrics were originally developed as tools to communicate to the general population the best and simplest way to identify individual foods with carbohydrates of high quality. However, there are no previous studies that have applied these different carbohydrate quality metrics to complete habitual dietary intake data and investigated their association with T2D risk. Therefore, the aim of this study was to investigate the association between these four different carbohydrate quality indices with various amounts of fibers and free sugars in relation to total carbohydrates and the risk of developing T2D in the Malmö Diet and Cancer Study (MDCS).

## 2. Materials and Methods

### 2.1. Study Design

The MDCS is a prospective cohort study that took place in the city of Malmö, Sweden. Between January 1991 and September 1996, recruitment and baseline examinations were implemented. A total of 74,138 people were recruited through personal invitation letters as well as advertisements in the media and posters in public places [19]. The study population consisted of men born between 1923 and 1945, as well as women born between 1923 and 1950, in Malmö, Sweden. The exclusion criteria were limited skills in the Swedish language, as well as mental disabilities that hindered the participants from answering the questionnaire at baseline. In total, 28,098 individuals took part in all baseline examinations. Additionally, individuals with a diabetes diagnosis at baseline were excluded (*n* = 1230), as well as individuals with missing data on leisure-time physical activity, smoking, or education (*n* = 246). A total of 26,622 participants, out of which 61.3% were women, were included in our study. 

### 2.2. Examinations at Baseline

The participants visited the study center to receive information regarding the baseline examinations as well as the aim of the study. During the visit, the participants’ blood pressure, weight, height, waist circumference, and body fat percentage (using bioimpedance) were measured. They also filled out a self-administered questionnaire regarding their education level, physical activity, occupation, alcohol consumption, current health status, and medical history, as well as which medications they were currently using and potential diseases among their close relatives. Lastly, the participants received instructions on how to fill out a food diary and a diet questionnaire. Two weeks later, at their second visit to the study center, the questionnaires were collected and double-checked for values that were missing. Furthermore, a diet interview was performed during the second visit. The MDCS was approved by the Regional Ethics Committee at Lund University (LU 51-90), and all participants provided their informed written consent during the first visit to the study center prior to the baseline examinations.

### 2.3. Dietary Assessment

During the baseline examinations, a dietary assessment was performed using a validated modified diet history method consisting of three parts. In the first part, information regarding the participants’ dietary habits was collected through a 7-day food diary tracking lunch, dinner, cold beverages, as well as nutritional supplements. To gain information about the intake of regularly consumed foods that were not included in the food diary, a 168-item food frequency questionnaire (FFQ) with information on portion sizes and frequencies was filled out. At their second visit, the dietary assessment was completed with a 60 min individual interview (changed to 45 min after 1 September 1994) to collect more detailed information regarding the participants’ food choices, portion sizes, and cooking methods. The information on food intake obtained from the food diary and the FFQ was compiled and documented as an average intake of grams/day of individual food items. Intakes of energy and nutrients (including total carbohydrates and fiber) were calculated using the food composition database by the Swedish National Food Agency consisting of approximately 1600 items of food, with recipes and food codes that were added for the MDCS. Free sugar intake had to be estimated by subtracting the estimated amount of sucrose and monosaccharides in fruit and vegetables (i.e., naturally occurring sugars) from the total intake of sucrose and monosaccharides (i.e., mainly fructose and glucose) [20]. In comparison to a reference method of 18-day weighed food records, the modified diet history method was found to have good ranking validity with energy-adjusted Pearson correlation coefficients for carbohydrate intake (0.66 and 0.70 for men and women, respectively), fiber intake (0.74 and 0.69), and sugar intake (0.60 and 0.74) [21].

### 2.4. Carbohydrate Quality Indices

Four indices were used in this study to assess the dietary carbohydrate quality of the participants:

1. A total of ≥1 g of fiber per 10 g of carbohydrates (10:1 carbohydrate:fiber);

2. A total of ≥1 g of fiber and <1 g of free sugars per 10 g of carbohydrates (10:1:1 carbohydrate:fiber:free sugar);

3. A total of ≥1 g of fiber and <2 g of free sugars per 10 g of carbohydrates (10:1:2 carbohydrate:fiber:free sugar);

4. A total of ≥1 g of fiber per 10 g of carbohydrates, and for each 1 g of fiber, <2 g of free sugars (10:1 carbohydrate:fiber and 1:2 fiber:free sugar, expressed as 10:1&1:2).

The 10:1 carbohydrate:fiber ratio, developed by the American Heart Association [18], is based on the carbohydrate-to-fiber ratio in whole wheat and recommends selecting foods with at least 1 g of fiber per 10 g of carbohydrates. The three additional indices, proposed by Liu et al. [18], were used to account for the dietary recommendations to limit free sugars and increase fiber intake. The 10:1:1 ratio was based on dietary recommendations for individuals to consume around 50% of energy from carbohydrates, 30 g of fiber per day, and <5% (<1 g per 10 g of carbohydrates) of energy from free sugars. The 10:1:2 ratio was based upon the same dietary recommendations as the 10:1:1 ratio, but the total energy intake from free sugars should be limited to <10% of the total energy intake (<2 g per 10 g of carbohydrates). Lastly, the 10:1&1:2 ratio, in addition to the 10:1 ratio for carbohydrate to fiber, limits every 1 g of fiber to 2 g of free sugars daily. The participants in the MDCS were categorized as whether they adhered to the indices or not (yes/no). 

### 2.5. Type 2 Diabetes Case Ascertainment

Participants with T2D were identified through various registers as well as a rescreening during the follow-up period until 31 December 2016. To be considered a case, a T2D diagnosis was required from a physician from the Swedish National Diabetes Register or the Regional Diabetes Registry 2000 of Scania, in line with the internationally established diagnostic criteria. The criteria included a fasting plasma glucose concentration of ≥7.0 mmol/L, or a fasting whole blood concentration of ≥6.1 mmol/L, measured at two separate events. Cases that had at least two Hemoglobin A1c (HbA1c) values of ≥6.0% were identified through the Malmö HbA1c Registry at Clinical Chemistry. Additional cases were identified through four national registries from the National Board of Health and Welfare in Sweden: the Swedish National Inpatient Registry, the Cause-of-Death Registry, and the Swedish Hospital-based outpatient care, as well as the Swedish Prescribed Drug Registry. Individuals with prevalent type 1 diabetes, secondary diabetes, latent autoimmune diabetes in adults, and other diabetes conditions were censored at the date of diagnosis. The participants contributed person-time from the date of enrollment in the cohort up until the date of their T2D diagnosis, death, migration from Sweden, or the end of the follow-up period.

### 2.6. Other Variables

Education was divided into five categories according to the highest level of education the participants had reported achieving. Their smoking status was divided into three categories (current smoker, former smoker, never smoked). Participants who reported no alcohol in the 7-day diary as well as reporting no alcohol during the previous year in the questionnaire were defined as zero consumers, and the rest were divided into sex-specific quintiles based on their alcohol intake with cut-offs as follows: 3.4, 9.1, 15.7, and 25.7 g/day for men and 0.9, 4.3, 8.1, and 14.0 g/day for women. To measure their leisure-time physical activity, a questionnaire compiled from the Minnesota Leisure-Time Physical Activity Questionnaire was used, and participants were asked to estimate how much time on average they spent during a week doing 17 different activities on average. The duration of time spent doing these activities was then multiplied by their respective metabolic equivalent intensity factor. A leisure-time physical activity score was created and divided into five categories based on the number of Metabolic Equivalent Task (MET)-hours per week. Seasons were divided into four categories: winter, spring, summer, and fall (January–March, April–June, July–September, and October–December, respectively). To account for a minor modification of coding routines of dietary data in September 1994 that reduced the interview time from 60 to 45 min, the diet method was added as a variable and was defined as old or new, i.e., before or after 1 September 1994. Participants who reported that they had made past dietary changes either because of illness or any other cause (yes or no) were identified from the questionnaire [22]. Potential misreporting of energy was identified by comparing the estimated energy expenditure (calculated from self-reported physical activity at leisure-time, at work, and household work; estimated sleep; and passive time) with the reported energy intake according to Black’s revised Goldberg method [23]. 

### 2.7. Statistical Analysis

All statistical analyses were performed using the software IBM SPSS Statistics version 27.0, IBM Corporation, with the *p*-value < 0.05 as statistically significant. The baseline characteristics of the population were analyzed for each index, with continuous variables (i.e., age, body mass index (BMI), carbohydrate intake, fiber intake, and free sugar intake) being described as mean (SD), and categorical variables (i.e., gender, dietary changes, educational level, smoking habits, alcohol consumption, and leisure-time physical activity) described as *N* (%). A Cox proportional hazards regression model analysis was used to study the risk of incident T2D in association with the various indices for carbohydrate quality (hazard ratio with a 95% confidence interval). In the basic model, we adjusted for age, sex, dietary method, season, and energy intake. In the multivariable model, we also adjusted for alcohol consumption, education, smoking habits, and leisure-time physical activity. In the third model, we further adjusted for BMI. A sensitivity analysis was carried out excluding participants who had changed their diet in the past and those with potential misreporting of energy intake. We also tested effect modification by sex and BMI by including a multiplicative factor (i.e., sex * index or BMI * index) in the model with all variables. 

## 3. Results

### 3.1. Baseline Characteristics 

A total of 26,622 individuals were included in the study population (61.3% women), with a mean age of 58.0 years (ranging from 44.5–73.6 years) and a mean BMI of 25.6 ± 3.9 kg/m^2^. The average energy intake was 2279 ± 653 kcal, and the mean carbohydrate, fiber, and free sugar intake was 244 ± 44.7 g (45.2 ± 6.05 E%), 20.1 ± 7.07 g, and 61.9 ± 33.3 g (11.0 ± 4.48 E%), respectively. The index with the highest adherence was the 10:1 carbohydrate:fiber index (19.9% of the population), while the lowest proportion of participants adhered to the 10:1:1 carbohydrate:fiber:free sugar index (1.9%). In addition, 12.1% of the population adhered to the 10:1:2 carbohydrate:fiber:free sugar index, and 14.6% of the population adhered to the 10:1&1:2 carbohydrate:fiber and fiber:free sugar index. Thus, there are 667 subjects who adhere to the 4th index (10:1&1:2 carbohydrate:fiber and fiber:free sugar) but not the 3rd index (10:1:2 carbohydrate:fiber:free sugar). This group consists of subjects with a too-high free sugar intake compared to carbohydrate intake, but with a free sugar intake within the limit if the free sugar intake is compared to fiber intake. In Table 1, the baseline characteristics of individuals within each carbohydrate quality index category are presented. In general, individuals who adhered to the indices of high carbohydrate quality were more likely to be women, be non-smokers, have a higher level of education, have reported a dietary change in the past, and underreport their energy intake. Alcohol consumption and leisure-time physical activity did not differ substantially between the categories. Mean carbohydrate intake (E%) and fiber intake were also rather similar across the indices and did not differ between those who adhered and did not adhere to the ratios. Fiber intake (g/1000 kcal) was significantly higher among those adhering to any of the indices. Free sugar intake (E%) was lowest in the group that adhered to the 10:1:1 carbohydrate:fiber:free sugar ratio, i.e., among participants who consumed ≥1 g of fiber and <1 g of free sugars per 10 g of carbohydrates. Individuals who adhered to the 10:1:1 index had a higher BMI on average.

### 3.2. Type 2 Diabetes Risk

The mean ± SD follow-up period was 18.4 ± 6.4 years. During the follow-up period, 4046 T2D cases (15.2%) were identified. Individuals who adhered to the 10:1:1 index ratio (1.9% of the population) had a higher risk of developing T2D in the multivariable model (hazard ratio (HR) 1.27; 95% CI 1.03–1.57) (Table 2). However, the association was attenuated, and non-significant, after adjusting for BMI (HR 1.08, 95% CI 0.87–1.33). There were no associations between the other indices (10:1, 10:1:2, and 10:1&1:2) and T2D risk (Table 2). Effect modification between sex and 10:1:1 was observed (*p*-interaction: 0.05), with a significant association with T2D risk for men (HR 1.43; 95% CI 1.00–2.05) but not for women (HR 0.97; 95% CI 0.75–1.27). No interaction with BMI was observed for any of the indices (*p*-interaction > 0.13). 

After carrying out a sensitivity analysis that excluded participants with past dietary changes and potential misreporters of energy (36% of the population), significant associations were found for the 10:1:2 and 10:1&1:2 indices. For individuals consuming ≥1 g of fiber and <2 g of free sugars per 10 g of carbohydrates (10:1:2 carbohydrate:fiber:free sugar ratio), an 18% lower risk (95% CI: 3–30%) was observed, and a 16% lower risk was observed among individuals consuming ≥1 g of fiber per 10 g of carbohydrates and, for each 1 g of fiber, <2 g of free sugars (10:1&1:2 carbohydrate:fiber and fiber:free sugar ratio). Similar risk ratios, although not statistically significant, were observed for the other indices after excluding participants with past dietary changes and potential misreporters of energy (Table 2). 

## 4. Discussion

The aim of this study was to investigate the association between carbohydrate quality, expressed as four different indices regarding fiber and free sugar intake based on recommendations from various health-related organizations, and the risk of developing T2D in the Malmö Diet Cancer cohort. After excluding individuals with past dietary changes, indicating unstable food habits, and individuals deemed to misreport their energy intake, a lower risk of developing T2D was observed for adherence across all indices, although statistically significant only for the 10:1:2 carbohydrate:fiber:free sugar and 10:1&1:2 carbohydrate:fiber and fiber:free sugar indices. Adherence to these two indices was characterized by relatively high fiber intake and moderate sugar intake. 

The abovementioned main results are generated from the analyses in which misreporters and diet changers were excluded. Participants reporting dietary change may have unstable food habits, and their reported dietary intake at baseline may therefore have less influence on the development of T2D. Excluding these subjects could consequently increase our possibility to capture associations between true long-term habitual dietary intakes and T2D risk. Indeed, these sensitivity analyses were demonstrated important as the model estimates differed compared to the full sample analyses for all four indices. For example, prior to these exclusions and additional adjustments for BMI, adherence to the index ratio representing an intake of ≥1 g of fiber and <1 g of free sugars per 10 g of carbohydrates (10:1:1 carbohydrate:fiber:free sugar) showed a slightly, but significantly, higher risk of developing T2D (HR = 1.27; 95% CI = 1.03–1.56). Participants adhering to this index had the highest mean BMI, and when adjusting for BMI (which is a potential mediating factor), the risk was attenuated and non-significant. When participants with past dietary changes and misreporting of energy intake were excluded, the risk ratio was 0.79 (0.51–1.23). Only 1.9% of the population adhered to the 10:1:1 index, which is the ratio where free sugar intake was most restricted. We have previously found that individuals with very low intakes of free sugar have a higher risk of mortality compared to those with a more moderate free sugar intake [20]. The differences between regression model estimates suggest that the sensitivity analysis and BMI adjustment likely have dealt with bias from reverse causation. In addition, due to the low adherence, it is difficult to evaluate this index in our population.

No previous studies have been performed regarding the associations between the four different carbohydrate indices and T2D incidence. However, there are previous studies that have assessed carbohydrate quality using other metrics in association with T2D risk. In a study of 70,025 women from the Nurses’ Health Study, a diet high in carbohydrates was associated with a slightly lower risk of developing T2D, as long as the diet was also high in cereal fiber and/or total fiber [24]. In a cross-sectional study by Fontanelli et al., among 3194 residents of São Paulo, the ≤10:1 carbohydrate:fiber ratio characterized foods with a lower amount of total and added sugars, as well as saturated fat, and a higher amount of dietary fiber, protein, and various minerals [25]. A lower level of several cardiometabolic risk factors, such as insulin resistance, was observed in subjects who consumed more foods with a ≤10:1 carbohydrate:fiber ratio [25]. 

There are several strengths and weaknesses of this study. The participants filled out a self-assessed food diary and FFQ. Therefore, misreporting might be a potential shortcoming of this study, which could lead to misclassification of the exposure. Although the dietary history method uses self-reported data, it has been shown to be of high relative validity. It included three parts, which also assessed cooking methods and serving sizes. Importantly, the diet method was designed specifically to estimate fiber and fat intake. Another limitation is the fact that the dietary data were collected at one point in time, and participants might have altered their diet during the follow-up period. However, a sensitivity analysis was performed to exclude individuals with past dietary changes, so those included in these analyses probably followed a more stable dietary pattern. The risk of recall bias was limited by the prospective design. Even if adjustments were made for known risk factors and possible confounders, there is still a possibility that residual confounders that were not accounted for may have existed. The MDCS included men and women living in Malmö, Sweden, and had a mean age of 58 years. Further studies are necessary to assess the generalizability of the results to other populations and age groups. For example, food products that are sources of carbohydrates could differ between countries and age groups. However, considering the agreement between the findings of our study and earlier evidence that dietary fiber is beneficial and free sugars are possibly detrimental for T2D risk, these results could potentially be applicable to the general population. These results support the current recommendation of consuming high amounts of dietary fiber and reducing free sugar consumption to promote health and reduce disease risk. However, the potential gain of implementing these carbohydrate quality indices into dietary recommendations in addition to current separate recommendations for free sugar and fiber remains questionable and requires additional research. 

We have, in this study, examined indices that originally were designed for evaluating single food items and applied them to entire diet intake data. This is a new approach, and, therefore, these findings must be replicated in other cohorts, preferably using multiple dietary recalls. A larger study sample could also be beneficial to ensure that there is a larger number of individuals adhering to each index, which can create greater precision and make it easier to draw conclusions. 

## 5. Conclusions

This study aimed to determine if there was any association between carbohydrate quality and the risk of developing T2D in the MDCS. Although a slightly lower T2D risk was observed among those adhering to the carbohydrate quality indices, the results were only statistically significant for the 10:1:2 and 10:1&1:2 indices after excluding individuals with potentially unstable food habits. The results of this study support the current recommendation of consuming high amounts of dietary fiber and reducing free sugar consumption. 

## Figures and Tables

**Table 1 nutrients-15-03870-t001:** Baseline characteristics for participants within each carbohydrate quality index group.

	10:1 CHO:FI ^1^		10:1:1 CHO:FI:FS ^2^		10:1:2 CHO:FI:FS ^3^		10:1&1:2 CHO:FS&FS:FI ^4^	
	No (*n* = 21,328)	Yes (*n* = 5294)	No (*n* = 26,118)	Yes (*n* = 504)	No (*n* = 23,399)	Yes (*n* = 3223)	No (*n* = 22,732)	Yes (*n* = 3890)
Females (%)	12,238 (57.4)	4069 (76.7)	15,912 (60.9)	395 (78.4)	13,853 (59.2)	2454 (76.1)	2974 (58.7)	2974 (76.5)
	**Mean (SD)**							
Age (years)	58.0 (7.7)	57.8 (7.4)	58.0 (7.6)	56.0 (7.1)	58.1 (7.7)	57.4 (7.3)	58.0 (7.7)	57.6 (7.4)
Body mass index (kg/m^2^)	25.6 (3.9)	25.6 (4.0)	25.6 (3.9)	26.4 (4.5)	25.6 (3.9)	25.8 (4.0)	25.6 (3.9)	25.7 (4.0)
Total energy (kcal/day)	2345 (658)	2015 (559)	2290 (651)	1742 (527)	2324 (653)	1955 (555)	2333 (654)	1967 (551)
Carbohydrate intake (E%)	45.1 (5.9)	45.9 (6.5)	45.2 (6.0)	45.0 (7.4)	45.2 (6.0)	45.3 (6.6)	45.2 (6.0)	45.5 (6.6)
Fiber intake (g/1000 kcal)	8.4 (1.9)	13.0 (2.6)	9.2 (2.7)	13.9 (3.5)	8.8 (2.3)	13.1 (2.8)	8.7 (2.1)	13.2 (2.8)
Free sugar intake (E%)	11.7 (4.5)	8.3 (3.4)	11.2 (4.4)	3.2 (1.1)	11.7 (4.3)	6.3 (2.0)	11.7 (4.4)	6.9 (2.4)
	**N (%)**							
Education level								
Less than 9 years	9221 (43.2)	11,059 (34.7)	10,858 (41.6)	201 (39.9)	9881 (42.2)	1178 (36.5)	9667 (42.5)	1392 (35.8)
9 years	5470 (25.6)	1523 (28.8)	6878 (26.3)	115 (22.8)	6109 (26.1)	884 (27.4)	5918 (26.0)	1075 (27.6)
Upper secondary school	1914 (9.0)	465 (8.8)	2333 (8.9)	46 (9.1)	2111 (9.0)	268 (8.3)	2045 (9.0)	334 (8.6)
University, no degree	1806 (8.5)	540 (10.2)	2293 (8.8)	53 (10.5)	2030 (8.7)	316 (9.8)	1947 (8.6)	399 (10.3)
University degree	2917 (13.7)	928 (17.5)	3756 (14.4)	89 (17.7)	3268 (14.0)	577 (17.9)	3155 (13.9)	690 (17.7)
Smoking habits								
Current	6480 (30.4)	1077 (20.3)	7420 (28.4)	137 (27.2)	6866 (29.3)	691 (21.4)	6744 (29.7)	813 (20.9)
Former	6993 (32.8)	1941 (36.7)	8749 (33.5)	185 (36.7)	7726 (33.0)	1208 (37.5)	7483 (32.9)	1451 (37.3)
Never	7855 (36.8)	2276 (43.0)	9949 (38.1)	182 (36.1)	8807 (37.6)	1324 (41.1)	8505 (37.4)	1626 (41.8)
Alcohol consumption								
Zero consumers	1269 (5.9)	350 (6.6)	1572 (6.0)	47 (9.3)	1406 (6.0)	213 (6.6)	1344 (5.9)	275 (7.1)
Quintile 1	3931 (18.4)	974 (18.4)	4788 (18.3)	117 (23.2)	4332 (18.5)	573 (17.8)	4207 (18.5)	698 (17.9)
Quintile 2	4004 (18.8)	945 (17.9)	4876 (18.7)	73 (14.5)	4402 (18.8)	547 (17.0)	4263 (18.8)	686 (17.6)
Quintile 3	4025 (18.9)	1012 (19.1)	4946 (18.9)	91 (18.1)	4433 (18.9)	604 (18.7)	4309 (19.0)	728 (18.7)
Quintile 4	4021 (18.9)	1040 (19.6)	4975 (19.0)	86 (17.1)	4397 (18.8)	664 (20.6)	4289 (18.9)	772 (19.8)
Quintile 5	4078 (19.1)	973 (18.4)	4961 (19.0)	90 (17.9)	4429 (18.9)	622 (19.3)	4320 (19.0)	731 (18.8)
LTPA (METh/week)								
<7.5	2151 (10.1)	375 (7.1)	2484 (9.5)	42 (8.3)	2287 (9.8)	239 (7.4)	2239 (9.8)	287 (7.4)
7.5–15	3300 (15.5)	679 (12.8)	3904 (14.9)	75 (14.9)	3575 (15.3)	404 (12.5)	3488 (15.3)	491 (12.6)
15–25	4954 (23.2)	1170 (22.1)	6013 (23.0)	111 (22.0)	5394 (23.1)	730 (22.6)	5252 (23.1)	872 (22.4)
25–50	7607 (35.7)	2120 (40.0)	9543 (36.5)	184 (36.5)	8450 (36.1)	1277 (39.6)	8186 (36.0)	1541 (39.6)
>50	3316 (15.5)	950 (17.9)	4174 (16.0)	92 (18.3)	3693 (15.8)	573 (17.8)	3567 (15.7)	699 (18.0)
Past dietary changes	4037 (18.9)	1900 (35.9)	5720 (21.9)	217 (43.1)	4747 (20.3)	1190 (36.9)	4475 (19.7)	1462 (37.6)
Energy reporting								
Under	2760 (12.9)	1284 (24.3)	3827 (14.7)	217 (43.1)	3130 (13.4)	914 (28.4)	2972 (13.1)	1072 (27.6)
Adequate	17,815 (83.5)	3943 (74.5)	21,473 (82.2)	285 (56.5)	19,484 (83.3)	2274 (70.6)	18,980 (83.5)	2778 (71.4)
Over	753 (3.5)	67 (1.3)	818 (3.1)	2 (0.4)	785 (3.4)	35 (1.1)	780 (3.4)	40 (1.0)

^1^ A total of > 1g fiber per 10 g of carbohydrates. ^2^ A total of ≥1 g fiber and <1 g free sugars per 10 g of carbohydrates. ^3^ A total of ≥1 g fiber and <2 g free sugars per 10 g of carbohydrates. ^4^ A total of ≥1 g fiber per 10 g of carbohydrates, and with each 1 g of fiber, <2 g free sugars. Abbreviations: CHO, carbohydrates; FI, fiber; FS, free sugar; SD, Standard deviation; LTPA, Leisure-time physical activity; METh, Metabolic equivalent task hour.

**Table 2 nutrients-15-03870-t002:** Association (HR and 95% CI) between the four carbohydrate quality indices and risk of type 2 diabetes in the Malmö Diet and Cancer cohort.

	10:1 CHO:FI ^1^		10:1:1 CHO:FI:FS ^2^		10:1:2 CHO:FI:FS ^3^		10:1&1:2 CHO:FS&FS:FI ^4^	
	No	Yes	No	Yes	No	Yes	No	Yes
**Full sample**								
N/cases	21,328/3290	5294/756	26,118/3956	504/90	23,399/3587	3223/459	22,732/3501	3890/545
Basic model ^5^	1.00	0.94 (0.86–1.02)	1.00	1.26 (1.02–1.56)	1.00	0.94 (0.85–1.03)	1.00	0.91 (0.83–1.00)
Multivariable model ^6^	1.00	1.00 (0.92–1.08)	1.00	1.27 (1.03–1.56)	1.00	0.99 (0.89–1.09)	1.00	0.96 (0.88–1.06)
Multivariable model + BMI ^7^	1.00	0.99 (0.91–1.08)	1.00	1.08 (0.87–1.34)	1.00	0.95 (0.86–1.05)	1.00	0.93 (0.85–1.02)
**Excluding diet changers and misreporters**								
N/cases	14,512/2055	2580/297	16,918/2332	174/20	15,610/2191	1482/161	15,312/2154	1780/189
Multivariable model + BMI ^7^	1.00	0.89 (0.79–1.01)	1.00	0.79 (0.51–1.23)	1.00	0.82 (0.70–0.97)	1.00	0.84 (0.72–0.97)

^1^ A total of > 1g fiber per 10 g of carbohydrates. ^2^ A total of ≥1 g fiber and <1 g free sugars per 10 g of carbohydrates. ^3^ A total of ≥1 g fiber and <2 g free sugars per 10 g of carbohydrates. ^4^ A total of ≥1 g fiber per 10 g of carbohydrates, and with each 1 g of fiber, <2 g free sugars. ^5^ Adjusted for age, sex, dietary method, season, and energy intake. ^6^ Adjusted for age, sex, dietary method, season, energy intake, alcohol consumption, smoking habits, leisure-time physical activity, and education. ^7^ Adjusted for age, sex, dietary method, season, energy intake, alcohol consumption, smoking habits, leisure-time physical activity, education, and BMI. Abbreviations: CHO, carbohydrates; FI, fiber; FS, free sugar; BMI, body mass index.

## Data Availability

The dataset presented in this article is not readily available because of ethical and legal restrictions. The data are available after application to the MDC Steering Committee (http://malmo-kohorter.lu.se/malmo-cohorts, accessed on 4 September 2023).

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
