# Peer review of "Comparisons of Different Carbohydrate Quality Indices for Risk of Type 2 Diabetes in the Malmö Diet and Cancer Study"

_nutrients, 2023, doi:10.3390/nu15183870_

Round 1
Reviewer 1 Report
The study aims to explore the relationship between carbohydrate quality indices based on fiber and free sugar intake and the risk of developing Type 2 diabetes. It employs a prospective cohort design and includes a sizable sample of participants. Four indices are investigated, and the results suggest an association between adherence to certain indices and a reduced risk of Type 2 diabetes. The findings contribute to the understanding of dietary factors influencing diabetes risk and underscore the importance of high fiber intake and reduced free sugar consumption.
The study design, data collection, and statistical analyses are robust and well-described. The use of a large prospective cohort enhances the study's credibility and supports the validity of the findings. The study addresses a gap in the literature by evaluating carbohydrate quality indices considering fiber and free sugar intake, providing a fresh perspective on their relationship with Type 2 diabetes risk. The findings have clear implications for dietary recommendations and public health strategies aimed at preventing Type 2 diabetes.
Suggestions for Improvement:
1. While the authors conducted a sensitivity analysis to exclude participants with past dietary changes and potential misreporting, further elaboration on the rationale for this exclusion and its potential impact on the results would enhance the manuscript's clarity.
- The authors acknowledge potential limitations in generalizing the findings to other populations and ages due to the cohort's characteristics. Further discussion on the implications and applicability of the findings to broader contexts would strengthen the discussion section.
Author Response
The study aims to explore the relationship between carbohydrate quality indices based on fiber and free sugar intake and the risk of developing Type 2 diabetes. It employs a prospective cohort design and includes a sizable sample of participants. Four indices are investigated, and the results suggest an association between adherence to certain indices and a reduced risk of Type 2 diabetes. The findings contribute to the understanding of dietary factors influencing diabetes risk and underscore the importance of high fiber intake and reduced free sugar consumption.
The study design, data collection, and statistical analyses are robust and well-described. The use of a large prospective cohort enhances the study's credibility and supports the validity of the findings. The study addresses a gap in the literature by evaluating carbohydrate quality indices considering fiber and free sugar intake, providing a fresh perspective on their relationship with Type 2 diabetes risk. The findings have clear implications for dietary recommendations and public health strategies aimed at preventing Type 2 diabetes.
Suggestions for Improvement:
- While the authors conducted a sensitivity analysis to exclude participants with past dietary changes and potential misreporting, further elaboration on the rationale for this exclusion and its potential impact on the results would enhance the manuscript's clarity.
ANSWER: We have in the revised version further elaborated on the rational and the potential impact (line 286-306).
“The abovementioned main results are generated from the analyses in which misreporters and diet changers were excluded. Participants reporting dietary change may have unstable food habits, and their reported dietary intake at baseline may therefore have less influence on the development of T2D. Excluding these subjects could consequently increase our possibility to capture associations between true long-term habitual dietary intakes and T2D risk. Indeed, these sensitivity analyses were demonstrated important as the model estimates differed compared to the full sample analyses for all four indices. For example, prior to these exclusions and additional adjustment for BMI, adherence to the index ratio representing an intake of >1 g of fiber and <1 g of free sugars per 10 g of carbohydrates (10:1:1 carbohydrate:fiber:free sugar), showed a slight, but significant, higher risk of developing T2D (HR=1.27; 95% CI=1.03-1.56). Participants adhering to this index had the highest mean BMI, and when adjusting for BMI (which is a potential mediating factor), the risk was attenuated and non-significant. When excluded participants with past dietary changes and misreporting of energy intake, the risk ratio was 0.79 (0.51-1.23). Only 1.9% of the population adhered to the 10:1:1 index which is the ratio where free sugar intake was most restricted. We have previously found that individuals with very low intakes of free sugar have a higher risk of mortality compared to those with more moderate free sugar intake [20]. The differences between regression model estimates suggest that the sensitivity analysis and BMI adjustment likely have dealt with bias from reverse causation.”
- The authors acknowledge potential limitations in generalizing the findings to other populations and ages due to the cohort's characteristics. Further discussion on the implications and applicability of the findings to broader contexts would strengthen the discussion section.
ANSWER: We have in the revised version added some additional discussion on the implication (line 346-351).
“These results support the current recommendation of consuming high amounts of dietary fiber and reducing free sugar consumption to promote health and reduce disease risk. However, the potential gain of implementing these carbohydrate quality indices into dietary recommendations in addition to current separate recommendations for free sugar and fiber remains questionable and requires additional research.”
Reviewer 2 Report
Review comments:
The objective of this article was to investigate the potential association between carbohydrate quality and the risk of developing type 2 diabetes in the Malmö Diet and Cancer Study (MDCS). It has a clear idea and a rigorous structure. If the author can improve the following question, the article will be more complete.
In the part of “Baseline characteristics”, line 226-227, you mentioned "Alcohol consumption and leisure-time physical activity did not differ between the categories", but the statement appears to lack explanatory support from Table 1. The provided results do not sufficiently indicate whether the observed differences between the groups were statistically significant, suggesting that including p-values in Table 1 would enhance reader comprehension of the findings.
Several grammatical errors, namely
1、Change "prevented the participants from answering the questionnaire at baseline." in lines 87 to "hindered participants from responding to the baseline questionnaire." to enhance the structural flow.
2、Add an apostrophe to the possessive form of the noun after "participants" in line 111 to indicate ownership.
3、Change "the recommendation to select foods" in line 141 to "recommends selecting foods" so that the verb agrees with the subject.
4、Replace line 225 "have a higher education" with "have a higher level of education" to use a more accurate expression.
5、Some colloquial or less formal expressions in the conclusion should be changed into words that are more suitable for academic writing, such as "find out" to "determine" and "seen" to "observed".
Author Response
The objective of this article was to investigate the potential association between carbohydrate quality and the risk of developing type 2 diabetes in the Malmö Diet and Cancer Study (MDCS). It has a clear idea and a rigorous structure. If the author can improve the following question, the article will be more complete.
In the part of “Baseline characteristics”, line 226-227, you mentioned "Alcohol consumption and leisure-time physical activity did not differ between the categories", but the statement appears to lack explanatory support from Table 1. The provided results do not sufficiently indicate whether the observed differences between the groups were statistically significant, suggesting that including p-values in Table 1 would enhance reader comprehension of the findings.
Answer: According to the Strengthening the Reporting of Observational Studies in Epidemiology (STROBE) guidelines, significance tests should be avoided in descriptive tables (Vandenbroucke et al Ann Int Med 2007). We therefore avoid presenting p-values for table 1.
We modified the sentence. It now reads: “Alcohol consumption and leisure-time physical activity did not differ substantially between the categories.”
Several grammatical errors, namely
1、Change "prevented the participants from answering the questionnaire at baseline." in lines 87 to "hindered participants from responding to the baseline questionnaire." to enhance the structural flow.
2、Add an apostrophe to the possessive form of the noun after "participants" in line 111 to indicate ownership.
3、Change "the recommendation to select foods" in line 141 to "recommends selecting foods" so that the verb agrees with the subject.
4、Replace line 225 "have a higher education" with "have a higher level of education" to use a more accurate expression.
5、Some colloquial or less formal expressions in the conclusion should be changed into words that are more suitable for academic writing, such as "find out" to "determine" and "seen" to "observed".
ANSWER: All these grammatical errors have been corrected.
Reviewer 3 Report
The Authors presented very interesting and detailed analysis about the impact of carbohydrate quality on risk of T2DM in the Malmö Diet and Cancer Study. The study is well designed and the discussion is concise and clearly described. However, some significant remarks must be addressed, regarding the results:
1. please provide p-values in both tables, especially in Table 2 to prove which regression models are statistically significant;
2. are there any significant correlations between carbohydrate quality and variables displayed in Table 1?
Author Response
The Authors presented very interesting and detailed analysis about the impact of carbohydrate quality on risk of T2DM in the Malmö Diet and Cancer Study. The study is well designed and the discussion is concise and clearly described. However, some significant remarks must be addressed, regarding the results:
- please provide p-values in both tables, especially in Table 2 to prove which regression models are statistically significant;
ANSWER: According to the Strengthening the Reporting of Observational Studies in Epidemiology (STROBE) guidelines, significance tests should be avoided in descriptive tables (Vandenbroucke et al Ann Int Med 2007). We therefore avoid presenting p-values for table 1.
For table 2, we provide 95% CI, and therefore p-values are superfluous.
- are there any significant correlations between carbohydrate quality and variables displayed in Table 1?
ANSWER: According to the Strengthening the Reporting of Observational Studies in Epidemiology (STROBE) guidelines, significance tests should be avoided in descriptive tables (Vandenbroucke et al Ann Int Med 2007). We therefore avoid presenting p-values for table 1.